# The Wildfire Dataset: Enhancing Deep Learning-Based Forest Fire Detection with a Diverse Evolving Open-Source Dataset Focused on Data Representativeness and a Novel Multi-Task Learning Approach

**Ismail El-Madafri** [1,*] , **Marta Peña** [2] **and Noelia Olmedo-Torre** [1]

1 Department of Graphic and Design Engineering, Universitat Politècnica de Catalunya, C. Eduard Maristany 16, 08019 Barcelona, Spain; n.olmedo@upc.edu
2 Department of Mathematics, Universitat Politècnica de Catalunya, Av. Diagonal 647, 08028 Barcelona, Spain; marta.penya@upc.edu
* Correspondence: ismail.el.madafri@upc.edu

**Abstract:** This study explores the potential of RGB image data for forest fire detection using deep learning models, evaluating their advantages and limitations, and discussing potential integration within a multi-modal data context. The research introduces a uniquely comprehensive wildfire dataset, capturing a broad array of environmental conditions, forest types, geographical regions, and confounding elements, aiming to reduce high false alarm rates in fire detection systems. To ensure integrity, only public domain images were included, and a detailed description of the dataset's attributes, URL sources, and image resolutions is provided. The study also introduces a novel multi-task learning approach, integrating multi-class confounding elements within the framework. A pioneering strategy in the field of forest fire detection, this method aims to enhance the model's discriminatory ability and decrease false positives. When tested against the wildfire dataset, the multi-task learning approach demonstrated significantly superior performance in key metrics and lower false alarm rates compared to traditional binary classification methods. This emphasizes the effectiveness of the proposed methodology and the potential to address confounding elements. Recognizing the need for practical solutions, the study stresses the importance of future work to increase the representativeness of training and testing datasets. The evolving and publicly available wildfire dataset is anticipated to inspire innovative solutions, marking a substantial contribution to the field.

**Keywords:** forest fire detection; remote sensing forest monitoring; deep learning; dataset representativeness; practical implementation; false positives; multi-task learning; multi-class classification; benchmark dataset; open source

## 1. Introduction

As the impact of climate change continues to intensify worldwide, the frequency and severity of wildfires have noticeably increased, posing substantial risks to ecosystems and human communities alike [1]. Efficient and effective early detection systems are vital for mitigating the destructive consequences of such events. Aligning with the United Nations' Sustainable Development Goals (SDGs)—specifically SDG 13, targeting urgent action to combat climate change and its impacts, and SDG 15, focusing on the sustainable use of terrestrial ecosystems and halting deforestation—these systems represent an essential move towards a resilient future.

Forest fires are multifaceted phenomena, influenced by various environmental and contextual factors. Designing accurate detection systems thus presents substantial challenges. The integration of deep learning (DL) techniques, especially within the field of

computer vision, has yielded impressive state-of-the-art results in addressing these complexities [2]. These models, primarily convolutional neural networks (CNNs), are capable of distinguishing between normal forest conditions and different stages of forest fires, and even recognizing the early indicators of wildfires [3,4]. However, their efficacy is heavily reliant on the quality, diversity, and relevance of the training data. Substandard or insufficiently diverse data can notably hinder these advanced algorithms' performance, possibly leading to increased false alarms or missed detections [5].

Once trained and evaluated, these models can be incorporated into various surveillance systems, such as satellites, drones, or ground-based platforms, for real-time image analysis [4]. When they detect visual features indicative of potential forest fires, they can promptly alert authorities, significantly enhancing response times and reducing wildfire's destructive effects. The continual improvement of these algorithms, facilitated by new data, ensures adaptability to shifting conditions and evolving wildfire patterns in the context of global climate change.

From a practical implementation standpoint, adopting a multi-modal data approach stands out as a highly robust solution for forest fire detection using DL [4]. This strategy leverages the inherent strengths of various data types, such as red-green-blue (RGB) data, thermal data, hyperspectral data, laser infrared technology, and meteorological data. Each of these offers unique insights and complementary information about the environment under observation. By combining these data types, the system can counterbalance the individual limitations of each, yielding a more accurate and comprehensive detection solution [6].

In the specific context of RGB data, relying exclusively on them can lead to a higher incidence of both false positives and negatives [4]. This phenomenon stems from the inherent limitations in RGB data's ability to consistently identify features indicative of forest fires. Characteristics such as color, shape, and texture can vary significantly across different fire scenes, leading to a lack of standardized identification markers and thereby complicating detection [7]. Additionally, confounding environmental factors such as clouds, fog, sunlight reflection, and low-altitude cloud cover can imitate visual features of fire or smoke, further affecting machine learning (ML) model performance [4] (see Figure 1).

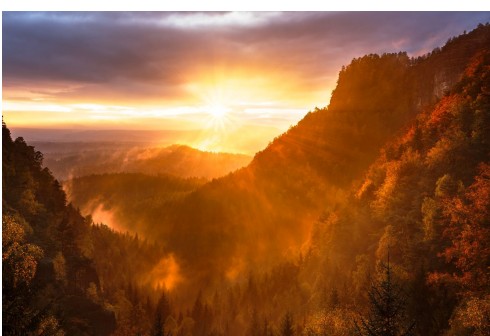 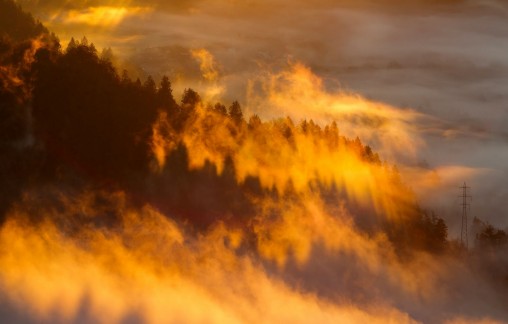

**Figure 1.** Images from the introduced dataset depicting visual features that may be misinterpreted by the model as a forest fire scene.

However, despite these challenges, RGB data remain vital to the development of DL-based forest fire early warning systems. Its broad availability, high resolution, and ease of human interpretation render them an indispensable element [5]. The widespread use of RGB imaging devices and their potential for frequent data collection ensure extensive spatial and temporal coverage. Consequently, RGB data stand as a robust primary source that, when thoughtfully combined with other data types, can markedly improve the fire detection system's overall performance. The strategic utilization of RGB data within a multi-modal data fusion framework presents a particularly promising approach to forest fire detection. While RGB data have their limitations, the unique attributes they possess make them a key component in the data fusion puzzle.

Maximizing the utilization of RGB data is appealing due to their near-ubiquitous availability and relative affordability. The high-frequency capture of RGB data allows for timely response to potential fire events, and the immediate mobilization of firefighting resources, a crucial factor in minimizing the devastating impact of forest fires. Moreover, RGB images offer a human-readable format that facilitates communication and coordination among various stakeholders, including decision-makers, firefighting teams, and the public [3]. Additionally, most state-of-the-art CNNs are pre-trained on RGB data, enabling transfer learning to reduce processing cost and training time, both of which are important aspects for the use of edge devices with limited capacity [8]. While the robustness of a multi-modal system is enhanced through the inclusion of other data types such as thermal, hyperspectral, laser infrared technology, and weather data to cite a few, the centrality of RGB data in this fusion approach is key.

Building on the integral role of RGB data in forest fire detection context, numerous strategies have been proposed to enhance the performance of DL models specifically trained on this data type. Addressing the unique challenges associated with forest fire detection in RGB images, these approaches span a broad spectrum of techniques. Data augmentation strategies have been deployed to enrich the diversity and representation of training data, bolstering the model's capacity to generalize across varying conditions [9]. Further improvements have been attained through transfer learning. This approach repurposes models pre-trained on large, diverse datasets for the specific task of forest fire detection, thus harnessing their robustness in handling confounding elements [10]. To refine the differentiation between fire and non-fire phenomena, multi-class classification schemes have been introduced. Preprocessing techniques play a vital role as well. Techniques such as image enhancement, background subtraction, and color space transformations have been employed to accentuate smoke and fire features in the images, thereby boosting the model's capacity to distinguish these phenomena. Researchers have also ventured into more advanced DL architectures, including recurrent neural networks (RNNs), transformer models, and attention mechanisms [11]. These methodologies have been proven to be valuable in prioritizing more relevant and distinctive regions within images, thus enhancing the overall model performance. On the other hand, saliency detection has shown promise for DL-based wildfire identification [7]. By directing the model's focus towards the most relevant and distinctive regions within images, this approach mitigates the impact of confounding elements on classification accuracy, and further improves model performance.

The development of algorithms for forest fire detection requires the efficient extraction of intricate visual features from varied data sources. However, an equally vital, yet often overlooked aspect, is the quality and representativeness of the data used [12]. While the drive to improve algorithmic performance continues, the necessity of capturing the complex and dynamic environmental conditions of real-world forests in training datasets cannot be understated. Unfortunately, this essential component has not received substantial attention in the existing research, leading to a critical gap in the field.

Detecting forest fires, whether from aerial sources or ground-based platforms, is a complicated process. It is fraught with variability due to environmental factors such as weather conditions, terrain, and vegetation, as well as caption-related aspects such as image resolution and angle of capture [4]. Datasets that fail to encompass this wide array of variables may inadvertently lead to models that cannot generalize to new, unseen scenarios. The consequences of this shortfall can be severe, culminating in suboptimal performance when applied to real-world situations, and possibly failing to detect or falsely detecting fire events.

Furthermore, a thorough review of existing literature on forest fire detection using RGB images uncovers a noticeable deficiency in understanding the confounding elements and challenges that contribute to high false alarm rates [13]. Although many studies acknowledge the connection between these factors and false alarms, they tend to explore the issue through a restricted set of examples. An in-depth examination that fully comprehends the nuances of these contributing elements is conspicuously absent. This lack of comprehensive

exploration leaves a significant opportunity for future research to bridge this knowledge gap, enhancing our ability to develop more precise and reliable fire detection systems.

Despite the significant strides made in the realm of ML-based forest fire detection, certain practical challenges remain unaddressed. The availability of publicly accessible datasets for developing and testing models is notably limited [12]. This assertion is further corroborated by recent review articles in the field [3,5,13]. This deficit impedes the establishment of a well-grounded benchmark, critical for enabling consistent evaluation and comparison of different forest fire detection models [5]. The absence of such standardized benchmarks hinders progress and validation of innovative techniques in this field [12]. A major contributing factor to this issue is the scarcity and limited accessibility of RGB wildfire data [5]. Predominantly sourced from fire surveillance cameras and drones, these images are often subject to permissions from local authorities. Moreover, capturing images of forest fires is not only hazardous for personnel involved but also difficult due to the unpredictable nature of wildfires. Coupled with the fact that forest fires are less common than other types of fires, acquiring adequate samples for research purposes becomes a fundamental task [3].

While there is a growing interest in leveraging RGB images for wildfire detection, the reality is that most studies either do not share their datasets or rely on private datasets, often fraught with various data-related issues [5]. Some research efforts, as documented in [14], have tried to tackle the problem by collecting images with visual elements similar to wildfires, such as fog, clouds, sunlight reflections, and sunsets, from non-specific wildfire datasets to reduce false positive rates. In another instance, researchers [9] made use of publicly accessible images from the Portuguese Firefighters Portal Database, a dedicated media outlet supporting Portuguese firefighters [15]. While this source provides a wealth of images from various fire incidents throughout Portugal, accessing these images for research is a laborious process. Each image must be downloaded individually, and media outlet logos need to be cropped before use. These hurdles underline the lack of standardized, diverse datasets in the field, complicating the task of comparing the efficacy of different methodologies and building on the reported results in the literature. In the same sense, it is notable that some studies have utilized video datasets for flame detection, though only a handful have been specifically curated for forest fires. Early wildfire warning systems necessitate datasets that encapsulate not just the flames, but also the smoke characteristics [14]. Predominantly, these training datasets consist of video frames, which are often plagued by an abundance of duplicate images. Such redundancy potentially undermines the generalizability of the models trained on these datasets [9]. Seeking alternatives to these constraints, researchers have ventured into the realm of synthetic data generation using generative adversarial networks (GANs) [14]. These networks can create additional training samples, infuse diversity, and facilitate training under controlled conditions. On the other hand, reference [16] exemplifies an alternative approach that targets some of the noted constraints by employing a multi-task learning (MTL)-based forest fire detection model (MTL-FFDet). The model was developed with three distinct tasks: the detection task, the segmentation task, and the classification task. This innovative approach shares the feature extraction module across all tasks, thereby enhancing feature extraction ability and reducing the number of false and missed detections. Furthermore, the introduction of a novel joint multi-task non-maximum suppression (NMS) processing algorithm seeks to leverage the benefits of each task to maximize detection accuracy.

Finally, when considering the practical applications of fire detection models in real-world forest environments, a comprehensive approach is warranted. This approach would ideally integrate video and image datasets with synthetic data generated by GANs. Video datasets bring to the table valuable temporal information, facilitating the monitoring of fires over time and encapsulating the dynamic nature of wildfires. On the other hand, image datasets collected from the Internet contribute a diverse set of samples, originating from varied sources and depicting fires under a wide spectrum of conditions and contexts.

The incorporation of synthetic data enhances the breadth of the training set, potentially improving the robustness and generalizability of the models.

In this study, we present a meticulously curated and diverse image dataset containing 2700 RGB instances, designed to serve as a benchmark for future forest fire detection research. The dataset is structured into two main categories (fire vs. nofire) and further divided into five subclasses, introducing a novel and comprehensive scope. It encompasses a wide array of environmental conditions, forest types, geographical regions, and confounding elements, all aimed at addressing the pervasive issue of high false alarm rates in DL-based fire detection systems.

Considering the notable scarcity of inclusive RGB datasets in this area, our contribution represents a valuable resource for the research community. To ensure the dataset's integrity, we adhered strictly to legal compliance, including only images that belong to the public domain, and providing a detailed description of the dataset's characteristics. This approach offers researchers a complete understanding of its diversity and depth.

Our goal is to spur innovation and facilitate progress in forest fire detection; thus, the dataset will be made publicly available. Accompanying the dataset, a CSV document will also be released to the public. In this document, each image will be linked to its respective download URL for reference and will include details such as its resolution. Through these efforts, the study aims to fill current knowledge gaps and foster the development of more precise and reliable solutions in this vital field.

In addition to providing the dataset, our work includes a thorough examination of potential confounding elements that could challenge the performance of DL models. By exploring these factors (see Figure 1), we aim to deepen the understanding of the complexities involved in forest fire detection, further enhancing the applicability and efficacy of our research.

Through a meticulous examination of the dataset, and by compiling a list of challenging factors identified through both a comprehensive literature review and a visual inspection of the images, this study emphasizes the depth and relevance of the proposed dataset. The effectiveness of the new dataset, referred to as "wildfire", will be assessed by leveraging a combined dataset. This combined collection comprises several relevant, previously published datasets, amounting to a total of 36,775 images. More detailed information on the datasets included can be found in Table 1.

Subsequently, a DL model trained on the combined dataset will be evaluated using the wildfire dataset. This approach not only helps to confirm whether the initial list of confounding elements covers most of the challenges faced by a DL model in current literature but also assesses the model's performance on specific types of images not covered by the list. If the model performs poorly on certain image categories, it may be necessary to update the list to include additional confounding elements. Moreover, this method serves as a means to evaluate the quality and relevance of the collected dataset and demonstrates its potential to enhance the diverse set of confounding elements in forest fire detection. This analysis will help justify the need for such a dataset in the research community and establish the significance of the study's contribution to the field.

**Table 1.** Overview of the diverse sources from which images were procured to construct the combined dataset.

| Set | Class |
| --- | --- |
| The DeepFire dataset [17] | The DeepFire dataset contains images with a uniform resolution of $250 \times 250$ pixels, sourced from various websites from the Internet. The dataset includes a total of 1900 images, neatly balanced between two categories: 950 images depicting fire incidents and the remaining 950 representing nofire scenarios. |

**Table 1.** *Cont.*

| Set | Class |
| --- | --- |
| The FLAME dataset [18] | The FLAME (Fire Luminosity Airborne-Based Machine Learning Evaluation) dataset is a collection of aerial imagery gathered through visible spectrum sensors mounted on two types of drones, the Phantom 3 Professional and Matrice 200 [13]. For wildfire classification, it includes 48,010 RGB aerial images with a resolution of $254 \times 254$ pixels. These images are categorized into two classes: 17,855 images without fire and 30,155 images containing fire. |
| HPWREN/AI for Mankind [19] | A dataset curated by AI for Mankind using public domain images that are publicly accessible from the High Performance Wireless Research and Education Network (HPWREN). These visual data, once downloaded, underwent a careful annotation process aimed at classification. A total of 1340 images were bifurcated into two distinct categories, namely "smoke" with 717 instances and "no smoke" with 623 instances. |
| M. S. Prasad, "Forest Fire Images", Kaggle, 2022 [20] | The dataset was formed by consolidating and merging multiple smaller datasets found on the Internet. However, specific references or sources for these constituent datasets were not provided. This resulted in two main categories, 'Test Data' and 'Train Data'. 'Test Data' includes a total of 50 images, divided equally into fire and nofire subfolders, each containing 25 images. The training set is composed of 5000 images, with fire and nofire subfolders each containing 2500 images. |
| The High Performance Wireless Research and Education Network dataset [21] | The database consists of a collection of non-segmented wildfire smoke images. These photographs were captured from both ground and aerial perspectives. The database houses a total of 98 original images, all featuring wildfire smoke. |

Furthermore, the study proposes a novel approach centered on a multi-task learning [22] framework. In this method, a single base model is simultaneously trained to carry out two related tasks—binary classification (fire/smoke vs. no fire/smoke) and multi-class classification (different types of fire and confounding elements). The uniqueness of this approach lies in the concurrent consideration of auxiliary task classes of confounding elements during both processes. To the best of the authors' knowledge, this unique approach of integrating multi-class confounding elements in a multi-task learning framework is a first in the field of forest fire detection. This innovative dual-task training could potentially enhance the model's ability to distinguish between subtle differences among classes, thereby reducing the false positive rate.

The study also addresses the data imbalance problem evident in the proposed wildfire dataset, where 1047 instances form the fire class against 1653 instances in the nofire class. Care was taken to retain the inherent characteristics of the fire class, and the natural occurrence bias was preserved in the validation and test sets. Utilizing both the original and augmented training sets, the study implemented classical one-step and two-step MTL multi-class classification methods. These explored the subtle yet discernible influence of data balancing on key performance metrics such as accuracy, precision, recall, F1-score, and ROC-AUC score.

Section 2 will detail the key stages of the methodology and describe the materials utilized in this study.

## 2. Materials and Methods

### 2.1. Dataset Collection and Curation

There is an evident need in the literature for enhanced datasets that address the existing limitations and gaps in the field of forest fire detection. Many RGB datasets often lack the required variety and representation of real-world conditions, hindering the development and evaluation of robust detection models [13]. The novel wildfire dataset introduced in this study aims to tackle this issue by purposefully increasing the variability between samples and integrating confounding elements. This increased variability facilitates a more comprehensive evaluation of detection approaches across various real scenarios, contributing to the enhancement of DL-based forest fire detection methods and their practical implementation.

### 2.1.1. Image Collection and Dataset Structure Formation

A dataset of 2700 RGB aerial and ground-based images of forested areas was gathered from multiple online sources, including government agencies, Flickr, and Unsplash. This diverse dataset encompasses a broad spectrum of environmental conditions, forest types, geographical regions, and the highly dynamic characteristics of forest ecosystems and fire events. The image resolutions within the dataset are varied, as indicated by the following key statistics:

- Average resolution: $4057 \times 3155$ pixels
- Minimum resolution: $153 \times 206$ pixels
- Maximum resolution: $19{,}699 \times 8974$ pixels
- Standard deviation of resolution (width): 1867.47 pixels
- Standard deviation of resolution (height): 1388.60 pixels

These metrics highlight high-resolution imagery that captures detailed information favorable for precise analysis in deep learning applications for forest fire detection.

The dataset's images represent different real-world scales, mirroring the varied sources and contexts from which they were collected. This diversity in scale was carefully considered in the design of the deep learning experiments, with images resized to a consistent scale as detailed in Section 2.3. No preprocessing steps were applied to the images to ensure their versatility and usability in different contexts. However, preprocessing tailored to this study's objectives was applied in the context of the experiments, as explained in Section 2.3. This approach maintains the native resolution and natural variability of the images, enabling targeted adjustments that enrich each potential analysis.

Though the collection process may not be exhaustive, it supports the robustness and generalizability of the findings derived from the dataset's analysis to some degree. The authors acknowledge the need for continuous development and expansion of this image collection and have chosen to maintain the dataset as a dynamic entity. This evolving approach signifies that additional images, videos, and other relevant data types will be incrementally included, based on feedback and requirements from the dataset's users. Such adaptiveness not only broadens the dataset's scope and richness but ensures that it remains a relevant and comprehensive tool for current and future research.

The dataset was carefully divided into training (70%), validation (15%), and testing subsets (15%), with further categorization within the primary classes of fire and nofire. The training set, containing 1888 images, forms the foundation for model learning. It consists of 1157 nofire images and 731 fire images. These images are further divided into subclasses representing different aspects of wildfires and potential confounders. The validation directory, holding 402 images, is used to fine-tune model parameters and avoid overfitting, while the test directory contains 410 images for the final evaluation. The structure of these directories is consistent, ensuring an authentic assessment of predictive capabilities.

Finally, the images within each directory were randomized, enhancing a diverse representation across the dataset. (Refer to Table 2 for more details on the datasets' classes and subclasses instances distribution).

**Table 2.** Data distribution across dataset's subclasses.

| Set | Class | Subclass | Instances |
|---|---|---|---|
| Training | Nofire | Forested areas without confounding elements | 591 |
| | Nofire | Fire confounding elements | 254 |
| | Nofire | Smoke confounding elements | 330 |
| | Fire | Smoke from fires | 463 |
| | Fire | Both smoke and fire | 286 |
| Validation | Nofire | Forested areas without confounding elements | 127 |
| | Nofire | Fire confounding elements | 50 |
| | Nofire | Smoke confounding elements | 69 |
| | Fire | Smoke from fires | 99 |
| | Fire | Both smoke and fire | 57 |

**Table 2.** *Cont.*

| Set | Class | Subclass | Instances |
| --- | --- | --- | --- |
| | Nofire | Forested areas without confounding elements | 128 |
| | Nofire | Fire confounding elements | 52 |
| Test | Nofire | Smoke confounding elements | 71 |
| | Fire | Smoke from fires | 100 |
| | Fire | Both smoke and fire | 59 |

2.1.2. Capturing General Variability in the Dataset

To strengthen the representativeness and generalizability of the dataset, the image collection process aimed to capture a comprehensive range of variability from environmental and caption-related sources. The following parameters were meticulously considered:

**Environmental Variability**:

1.  **Topography**: Varied terrain features, including hills, valleys, slopes, and plateaus.
2.  **Canopy Density and Structure**: Distinct differences in tree density, height, branching patterns, and forest stratification.
3.  **Forest Types and Species Composition**: A variety of forest ecosystems encompassing diverse species, plant communities, and successional stages.
4.  **Ground Cover**: A wide range of ground cover types, such as grass, bare soil, water, rocks, and leaf litter.
5.  **Natural Components**: The presence of rivers, lakes, wetlands, and other natural landscape elements.
6.  **Human-made Objects**: Infrastructure, including roads, bridges, buildings, vehicles, power lines, and other anthropogenic features.
7.  **Weather Conditions**: Various atmospheric phenomena, such as fog, rain, snow, dust, and wind.
8.  **Foliage**: Seasonal and phenological changes in foliage, including leaves, flowers, fruits, and seed dispersal.
9.  **Sunlight**: Diverse sunlight exposure, shading patterns, and solar angles.
10. **Fire Characteristics**: Variability in fire size, shape, color, intensity, progression, and smoke plume dynamics.
11. **Smoke Dispersion**: The variability in smoke plume patterns due to wind speed, wind direction, and atmospheric stability.

**Caption-related Variability**:

1.  **Lighting Conditions**: Fluctuations in light and shadows resulting from clouds, time of day, and sun angle [9].
2.  **Image Resolution**: Varied levels of image detail, sharpness, and pixel density.
3.  **Altitude and Distance**: Diverse flying heights and distances from the forest or fire event, affecting image scale and detail [14].
4.  **Camera Angle and Orientation**: Variations in the camera angle relative to the subject, its orientation, and field of view.
5.  **Perspective**: A mix of top-down, oblique, and side-view angles in the images.
6.  **Platform Type**: Heterogeneous image sources, such as drones, planes, and helicopters.
7.  **Obstructions and Reflections**: The presence of objects or atmospheric conditions that may cast shadows, cause reflections, or influence image quality.
8.  **Image Compression**: The type and degree of image compression applied, which can potentially introduce artifacts or degrade image quality.
9.  **Camera Motion Blur**: The effect of camera motion or platform vibrations on image sharpness, which can occur during flights in turbulent conditions or at high speeds.

2.1.3. Rigorous Data Curation and Deduplication Process

Throughout the collection process, a meticulous approach was used to optimize the quality of the dataset, ensuring it provides valuable insights for the practical implementation of the models.

The dataset features fewer images of forested areas covered with snow, as these environments are less prone to forest fires.

In assembling the dataset, the authors placed particular emphasis on images that captured human interaction with forests. This approach included incorporating images of forested areas containing buildings, human settlements, roads, bridges, and other structures associated with human activities. This consideration is crucial, as the majority of forest fires are attributed to human actions.

To address potential duplication issues in the dataset, a deduplication method was employed. This process involved comparing the perceptual hashes of the images to eliminate instances of double counting caused by duplicates gathered from multiple data sources. To detect similar images with only slight, non-significant differences, several image comparison algorithms were implemented.

2.1.4. Data Sources, Licensing, and Permissions

The images for the wildfire dataset were collected from multiple online sources, including government agencies, Flickr, and Unsplash. To ensure compliance with intellectual property rights and usage permissions, the licensing and permissions for each source were meticulously verified before incorporating the images into the dataset. The sole licensing associated with the images within the dataset is the public domain dedication. This selection ensures legal compliance, precludes issues with incompatible licenses, and allows for the dataset to be freely shared within the research community. In addition, the dataset will be complemented by a Supplementary File. This file will serve as a reference guide, linking each image to its corresponding download URL. This approach ensures transparency and allows users to trace each image back to its original source, if needed.

*2.2. Model Selection: MobileNetV3*

Striking a balance between model performance and efficiency is a fundamental consideration in this study, given the significant computational demand and complexity involved in processing large-scale image datasets for forest fire detection. In this sense, to carry out the study effectively, the researchers chose MobileNetV3 [23] as a representative model for their study's experiments. Thanks to its compact and efficient design as a CNN, the model is ideal for image classification tasks. MobileNetV3 is a variant in the MobileNet series, combining the strengths of its predecessors, MobileNetV1 and V2, while integrating additional enhancements for improved performance. MobileNetV3's architecture is notably characterized by its use of depthwise separable convolutions, designed to minimize computational costs without significantly compromising model performance. This technique dissects the standard convolution operation into two separate layers: a depthwise convolution and a pointwise convolution. This separation effectively attenuates the computational load while preserving most of the network's representational power. Moreover, MobileNetV3 incorporates inverted residual blocks with linear bottlenecks, a technique that bolsters model capacity. These blocks, drawing inspiration from ResNet's architecture, consist of a series of layers where the input and output share the same dimensions, fostering easy information flow.

MobileNetV3, with its focus on efficient computation and enhanced accuracy compared to previous versions, emerges as an ideal model choice [23]. Its smaller parameter count facilitates more efficient training and deployment of the model, which proves advantageous in the practical implementation of forest fire detection systems. By using MobileNetV3 for their experiments, the researchers aim to show that promising performance can be attained without sacrificing efficiency. This balance is a critical factor in the development of practical and scalable forest fire detection solutions that rely on DL.

## 2.3. Training the MobileNetV3

The Keras framework with a TensorFlow backend and GPU support were used to compile the experiments. Training images are normalized to values between 0 and 1 by dividing each pixel by 255 and resized to the default size of MobileNetV3, which is $224 \times 224$ pixels. This resizing step ensures that the images are treated at a consistent scale, a key factor in the model's ability to detect forest fire conditions across the diverse dataset. To optimize the performance of the input pipeline for the training, validation, and test splits, the prefetch method was employed. This approach ensures that the processing unit is not waiting for data to be loaded while training or evaluating the model, leading to faster training times and improved overall performance. By using the prefetch method, the optimal buffer size for prefetching is automatically determined, greatly enhancing the efficiency of the data pipeline during both training and evaluation stages.

Stochastic optimization methods, including Stochastic Gradient Descent (SGD), Root Mean Square Propagation (RMSProp), and Adaptive Moment Estimation (Adam), are applied with a maximum of 100 epochs. Early stopping is configured with a patience of 5 and a minimum change in the loss of 1e-3. We determine the optimal Learning Rate by testing four different rates ($10^{-2}$, $10^{-3}$, $10^{-4}$, $10^{-5}$). A global average pooling layer is integrated to reduce the dimensionality of the output matrix from the convolutional layers, which is then flattened into a vector. This vector serves as an input for the fully connected prediction layer. A dropout regularization technique with a dropout rate of 0.2 is employed to enhance the model's generalization capability.

The model's performance is evaluated using a held-out validation dataset, and the combination of learning rate and optimizer that yields the highest accuracy is selected for testing on the test dataset.

## 2.4. Enhanced Detection with Multi-Task Learning Approach

This study introduces a novel approach, which includes the formulation of five distinct subclasses of the wildfire dataset based on the curated list of confounding elements. These classes are intended to ensure a balanced distribution of images depicting fire/smoke events and those not. The classes are as follows:

1. **Smoke from fires (subclass 1)**: This class encompasses images that illustrate smoke emissions from fires, without the apparent presence of flames.
2. **Both smoke and fire (subclass 2)**: This class includes images that exhibit both flames and smoke emissions from fires.
3. **Forested areas without confounding elements (subclass 3)**: Images devoid of any confounding elements, as per the defined list, are categorized under this class. They mainly represent typical forested areas.
4. **Fire confounding elements (subclass 4)**: This class comprises images that contain elements easily misconstrued as fire.
5. **Smoke confounding elements (subclass 5)**: Images that feature elements that may be misinterpreted as smoke fall under this class.

In the proposed approach, an auxiliary task of five-class classification is established alongside the primary task of binary classification (fire/smoke vs. nofire/nosmoke) within a MTL framework. A single base model is trained to handle both tasks concurrently. This strategy takes advantage of shared features between tasks, enhancing the model's ability to generalize and improve overall performance.

The efficacy of the hierarchical multi-class classification strategy will be assessed. This assessment involves comparison with a traditional one-step binary classification approach. In the one-step approach, a single model is directly trained to classify images into two categories: those showing a fire event (subclasses 1 and 2) and those not (subclasses 3, 4, and 5). This comparative analysis helps evaluate the potential benefits of implementing multi-class MTL-based classification and the importance of addressing confounding elements within the method.

Another significant aspect of the analysis is to gain insight into identifying common visual elements in the images that could resemble fire or smoke, such as sun glare, clouds, fog, or specific vegetation types. Training the model to recognize these elements could help reduce false positives, as the model learns to differentiate between actual fires/smokes and visually similar elements [22].

Finally, feature visualization techniques, specifically Gradient-weighted Class Activation Mapping (Grad-CAM) [24], are proposed. This approach helps understand which parts of the input images contribute the most to the model's predictions. Such understanding can assist in identifying the visual features shared by confounding elements and actual fires or smoke.

### 2.5. Addressing Confounding Elements

In assessing any alarm system, it is crucial to consider the presence of potential confounding elements that could lead to false alarms. The visual characteristics of certain elements may resemble those of smoke and forest fires, posing challenges in accurately distinguishing between images containing fire and those without. It is essential to understand and address these confounding factors in order to develop more accurate and reliable forest fire detection systems, minimizing the occurrence of false alarms and enhancing overall performance.

The process of addressing confounding elements starts by creating an initial list of challenging factors based on a comprehensive literature review. This review focuses on studies that have emphasized the connection between confounding elements and high false alarm rates in DL models trained on RGB forest fire detection. In addition to the literature review, an analysis of the initial wildfire dataset is conducted to identify any potentially relevant factors that may not have been addressed in existing research. Then, the list will be used to form five subclasses of nofire images that will be included for the multi-class classification problem for the remaining steps of the experimental setup.

The considered confounding elements are divided into specific subcategories, each presenting its unique set of challenges to DL models. The compiled list with descriptions is detailed below:

1. **Atmospheric Phenomena**: (a) Fog or mist: These can produce illusions of smoke due to their translucent and diffused appearances, leading to potential misclassifications [25]. (b) Low-altitude clouds: Their visual similarities to smoke plumes, particularly the gray or white clouds, pose challenges for models in distinguishing between them and smoke [7,19]. (c) Sunset: The angle and intensity of sunlight during sunset can produce shadows and bright spots, complicating the differentiation between fire and nofire elements.

2. **Vegetation and Seasonal Changes**: (a) Reddish/orange foliage: Some tree species display red and orange hues during specific seasons, which can be misconstrued as fire or embers in aerial images.

3. **Lighting and Reflections**: (a) Sunlight reflection on trees and water: Bright spots that mimic fire or smoke features can be produced when sunlight reflects off wet surfaces or water bodies [7]. (b) Shadow and lighting variations: Shadows that can be mistaken for smoke or fire may be created by changes in lighting conditions, such as those induced by clouds, time of day, or topography [9].

4. **Camera-Related Artifacts**: (a) Camera motion blur: Motion blur resulting from camera movement or platform vibrations can lead to the introduction of visual artifacts resembling smoke or fire.

5. **Visually Similar Objects [26] and Phenomena [14]**: This category encompasses any other objects or phenomena that visually resemble fire or smoke, presenting additional challenges for accurate classification.

As highlighted in Section 2.4, the focus of the study will include a multi-classification problem comprising five distinct classes. Two of these classes are specifically dedicated to confounding elements: one class embodies elements that mimic fire, while the other encom-

passes elements that imitate smoke. Following an analysis of the model's misclassifications, a deliberate emphasis will be placed on elements that are more frequently misclassified from the initial list. The goal of incorporating these classes dedicated to confounding elements is to augment the model's ability to distinguish between genuine fire and smoke characteristics and those that merely bear similarities. This, in turn, should enhance the precision and dependability of the forest fire detection system.

*2.6. The Data Balancing Problem*

In the realm of machine learning and data-driven models, dealing with an imbalanced dataset represents a sophisticated challenge. Balancing a dataset can mitigate biases and enhance model performance, particularly in cases where class distributions are inherently unequal, such as the observed imbalance between the fire and nofire classes in the present wildfire dataset [27]. With 1047 instances in the fire class against 1653 in the nofire class, this disparity in the collection process is not merely a statistical artifact; it reflects the actual occurrence bias existing in nature. The decision to employ data balancing techniques must therefore be handled with meticulous care. This includes maintaining the essential characteristics of each class without over-representation or artificial inflation that could distort the model's real-world applicability [28]. The choice of whether or not to balance, and how to do so, becomes a nuanced task that requires an intricate understanding of the data's structure, the model's purpose, and the underlying real-world dynamics of each forest. Since the main goal of the rest of the study is to assess the impact of novel strategies, such as the consideration of confounding element classes, the authors believed that failure to balance the classes might bias the model towards the majority class, limiting the robustness of the experiment's results. In the following sections, the specific approach to this multifaceted issue will be detailed, elucidating the careful considerations and methodologies employed to strike a delicate balance that, to a certain extent, preserves an authentic reflection of real-world scenarios.

To reduce the bias in the training process, the fire class in the training set was augmented to match the nofire class in terms of representation. Augmentation was proportionally distributed among the subclasses of the fire class, ensuring that each source image was utilized only once. This method minimized the risk of the model internalizing noise or peculiarities from augmented samples, allowing it to focus on the underlying patterns. As a result, 268 images were incorporated into the Smoke from fires subclass (subclass 1), and 158 into the Both smoke and fire subclass (subclass 2). Specific augmentation techniques were applied, including random rotations within a range of 40 degrees, width and height shifts of 20%, a shear range of 20%, zooming within a range of 20%, horizontal flipping, and using the 'nearest' method for filling in newly created pixels [29].

The process of balancing the classes within the training set was carefully designed to reduce the model's tendency to favor the majority class, potentially improving the ability to identify the minority class. By maintaining an even distribution, the training set aided the model in avoiding an overfit to specific categories, thus enhancing its ability to generalize.

In contrast, the authors decided to retain the natural distribution within the validation and test sets. The considerations guiding this decision included avoiding overfitting, preserving natural distribution in validation and test sets, and preventing data leakage. (Refer to Table 3 for more details on the datasets included).

The empirical comparison of the original and augmented training sets offered valuable insights into the influence of data balancing on model performance. These experiments underscored the importance of a nuanced approach to class balancing, reaffirming the methodology's alignment with best practices and its potential to support nuanced predictions in fire classification. As the primary objective of this empirical comparison is to evaluate the effect of data balancing on model performance, the same hyperparameters were maintained during the training of the models. This approach was taken with the intention of creating a more controlled comparison, where the only differing variable was the data itself.

**Table 3.** Distribution of original and augmented instances across the five subclasses before and after balancing the training set.

| Class | Subclass | Original Instances | Augmented Instances |
|---|---|---|---|
| Nofire | Forested areas without confounding elements | 846 | 847 |
| | Fire confounding elements | 338 | 336 |
| | Smoke confounding elements | 470 | 471 |
| Fire | Smoke from fires | 662 | 930 |
| | Both smoke and fire | 384 | 542 |
| Total | | 2700 | 3126 |

Beyond this assessment, all other experiments within the study utilized the augmented dataset, aligning with the broader methodology.

The augmentation process was designed to equalize the number of instances between the fire and nofire training set classes. By adding 268 images to the Smoke from fires subclass and 158 to the Both smoke and fire subclass through techniques such as random rotations, zooming, and flipping, both classes in the training set were brought to an equal count of 1157 instances each.

In the process of evaluating the impact of data balancing on model performance, it was observed that the differences in key performance metrics between models trained on the original and balanced datasets were not obviously substantial. Such variations raised questions concerning the stability of the observed differences, as minor fluctuations might result from random variations or noise inherent in the training process. Given these relatively narrow margins, a more nuanced and robust analysis was recognized as necessary. To this end, the model, for each method, was trained multiple times, utilizing both datasets, for a total of five iterations. The objective was to assess the stability and reliability of the results rather than to conduct formal statistical significance testing. Confidence intervals for the differences in performance metrics between the two methods were calculated using bootstrapping, a resampling technique that allows for robust statistical inference, particularly when dealing with small sample sizes (here, 5 runs). These intervals offer a range within which the true differences in the models' performances are likely to lie, providing insights into the statistical significance of the differences and contributing to a more comprehensive understanding of how the balancing through the augmentation of positive instances (fire images) affects various aspects of the models' behavior.

Further details of these experiments are provided in the results sections.

*2.7. Weighting of Confounding Elements Subclasses in Model Training*

In the multifaceted task of fire detection, the influence of confounding elements is a critical consideration. These factors, which differ in complexity and ambiguity, may have varied impacts on the model's performance. A specific class might be more prone to confusion with an actual fire or smoke event, thus requiring particular attention during the training phase. Additionally, the disparate real-world effects of these elements further justify differential weighting in their consideration. Imbalances within the dataset could also be tackled by varying weights to cultivate a more balanced learning environment.

To investigate these aspects, a systematic approach is deployed in the training process, where the weights of two subclasses, namely Fire confounding elements (subclass 4) and Smoke confounding elements (subclass 5), are manipulated. Starting with equal weights for both subclasses, the weights are methodically adjusted from 1 to 3, and specific combinations are tested to gauge the model's performance under different confounding circumstances. This strategy also sheds light on how the model's sensitivity to these elements can shape its overall efficacy.

After determining the optimal weights, a detailed examination is conducted to comprehend why this specific weighting is effective. This includes probing how these optimal weights sway the entire model performance and identifying the underlying mechanisms

that render them effective. By doing so, the study seeks to offer nuanced insights into the complex interplay of confounding factors in the realm of fire detection.

*2.8. Transfer Learning*

Different transfer learning [8] scenarios are evaluated, including training from scratch, fine-tuning, and feature extraction, to identify the most effective strategy for enhancing the model's classification.

*2.9. Performance Metrics*

The performance of the model is gauged using the four key elements of the confusion matrix: True Positives (TP), True Negatives (TN), False Negatives (FN), and False Positives (FP). TP and TN represent the accurate predictions of fire and nofire images, respectively, while FN and FP denote the instances where fire images and nofire images are incorrectly identified.

**Accuracy** is a measure of how often the model predicts correctly and is given by the ratio of correct predictions to total predictions.

$$\text{Accuracy} = (\text{TN} + \text{TP}) \Big/ (\text{TN} + \text{FN} + \text{FP} + \text{TP})$$

**Precision** quantifies the model's reliability when making positive predictions, defined as the ratio of correctly identified fire instances to all instances that the model labels as fire.

$$\text{Precision} = \text{TP} \Big/ (\text{FP} + \text{TP})$$

**Recall** (or sensitivity) expresses the proportion of actual fire images that are correctly identified by the model out of all actual fire images.

$$\text{Recall} = (\text{TN} + \text{TP}) \Big/ (\text{FN} + \text{TP})$$

The **F1-score** is a combined measure that reflects both precision and recall in a single metric, thus allowing for an overall evaluation of a model's predictive performance. This is in contrast to accuracy, which measures the overall rate of correct predictions, encompassing both fire and nofire predictions.

$$\text{F1\_score} = 2 \times (\text{Precision} \times \text{Recall}) \Big/ (\text{Precision} + \text{Recall})$$

The **ROC-AUC** (Receiver Operating Characteristic—Area Under Curve) score is a comprehensive metric that evaluates a model's ability to distinguish between classes. Unlike individual metrics such as accuracy, precision, or recall, ROC-AUC considers both the true positive rate (sensitivity) and the false positive rate (1-specificity) across different thresholds. It plots a curve (ROC curve) that represents these rates across all thresholds, and the AUC value calculates the area under this curve. A perfect classifier would have an ROC-AUC score of 1, while a completely random classifier would score 0.5. The ROC-AUC score provides insights into the model's discriminatory power, regardless of the specific threshold, making it a valuable metric for assessing a model's overall classification effectiveness.

## 3. Results

Upon applying the MobileNetV3 model to the wildfire dataset, we observed significant potential for enhancements in the model's capacity to detect forest fires. This included the skilled handling of complex variables known as confounding elements within the images.

This improvement was realized through a novel two-step multi-class classification strategy. The application of this approach strengthened the model's robustness, reduced false alarms, and increased its ability to adapt to different and often challenging situations specific to RGB forest fire detection.

As exposed previously, two distinct methods were investigated in this study: a one-step classification (referred to as Method 1) and a two-step MTL multi-class classification (referred to as Method 2). The experiments were conducted using both original and augmented training sets to thoroughly evaluate the impact of data balancing on the model's performance.

To facilitate a clear comparison between these two methods, their performance metrics were meticulously analyzed. The average values of these metrics after five independent runs, which gauge essential qualities such as accuracy and precision, are detailed in Table 4. The following provides an in-depth view of the comparative merits and limitations of each method, helping to illuminate their respective roles and potentials in wildfire detection.

**Table 4.** Model performance means metrics for Method 1 and Method 2 using the original and the balanced wildfire datasets.

| Method | Dataset | Accuracy | Precision | Recall | F1-Score | ROC-AUC |
|---|---|---|---|---|---|---|
| Method 1 | Original | 0.8405 | 0.8324 | 0.7799 | 0.8049 | 0.8397 |
| Method 1 | Balanced (Augmented) | 0.8156 | 0.7187 | 0.8667 | 0.7857 | 0.8258 |
| Method 2 | Original | 0.9073 | 0.8832 | 0.8855 | 0.8839 | 0.9053 |
| Method 2 | Balanced (Augmented) | 0.8766 | 0.7974 | 0.9171 | 0.8526 | 0.8842 |

The average values of key metrics after five independent runs.

### 3.1. Method 1: One-Step Classification

The one-step classification approach categorizes images directly into two distinct classes: fire and nofire events. Table 5 outlines the optimized hyperparameters for Method 1, with configurations derived from the validation dataset. Meanwhile, Table 6 presents the average values and standard deviation after five runs of performance metrics for Method 1, utilizing both the augmented and original wildfire datasets.

**Table 5.** Hyperparameter configuration for Method 1.

| Hyperparameter | Value |
|---|---|
| Learning Rate | $1 \times 10^{-4}$ |
| Optimizer | Adam |
| Batch Size | 32 |
| Epochs | 50 |
| Fine tune at the layer number | 80 |

**Table 6.** Performance metrics means for Method 1 using augmented and original wildfire datasets.

| Metrics | Augmented Wildfire Dataset | Original Wildfire Dataset |
|---|---|---|
| Primary Accuracy | Mean = 0.8156, Std = 0.0152 | Mean = 0.8405, Std = 0.0103 |
| Precision | Mean = 0.7187, Std = 0.0143 | Mean = 0.8324, Std = 0.0367 |
| Recall | Mean = 0.8667, Std = 0.0108 | Mean = 0.7799, Std = 0.0178 |
| F1-Score | Mean = 0.7857, Std = 0.0103 | Mean = 0.8049, Std = 0.0176 |
| ROC-AUC | Mean = 0.8258, Std = 0.0083 | Mean = 0.8397, Std = 0.0151 |

These results are based on five runs for each model, with bootstrapping used to calculate 95% confidence intervals for the differences in performance metrics between the two models.

Data Balancing Assessment for Method 1

The comparison between Method 1 trained on the original wildfire dataset and the augmented training dataset uncovers differences in performance across various metrics. The mean primary accuracy for the augmented dataset is 0.8156 (Std = 0.0142), whereas the original dataset results in a somewhat higher mean accuracy of 0.8405 (Std = 0.0094), with a significant difference detected within a 95% confidence interval of $[-0.03758, -0.01418]$. As expected, the augmented method exhibits significantly lower precision but a higher recall. Considering the potential bias in the original dataset related to the imbalance, the superior precision and overall accuracy might stem from this inherent data skewness. Conversely, the augmented dataset method, with its increased recall, seems to provide a more balanced perspective, enhancing sensitivity to fire instances.

The contrasts between these two methods also encompass the F1-Score, with the original method showing a marginally better performance, and the ROC-AUC metrics, where no significant difference was detected. These observations underline the complex interplay between different evaluation metrics and the necessity to ensure harmony between the model's goals and its assessment standards. Specifically, while the augmentation of positive fire instances improves sensitivity (recall), it seems to compromise precision and accuracy, though not significantly impacting the F1-Score and ROC-AUC. This intricate balance accentuates the significance of a well-thought-out data balance and augmentation scheme, demanding rigorous analysis and planning to attain the targeted model efficiency in forest fire detection.

The preference for one method over the other might thus hinge on particular goals and necessities in wildfire detection. The original dataset method might yield slightly superior overall results but could be influenced by bias due to the skewness in the training data. On the other hand, the augmented dataset method could grant benefits in areas such as sensitivity to fire occurrences, mirroring a more even-handed comprehension of the underlying data dynamics. This understanding emphasizes the imperative for an all-encompassing strategy in choosing and fine-tuning models, contemplating not merely performance indicators but also the innate attributes and conceivable biases of the datasets employed.

### 3.2. Method 2: Two-Step MTL Multi-Class Classification

In contrast to the first method, the second approach employs a hierarchical structure. Table 7 outlines the optimized hyperparameters for Method 2, with configurations derived from the validation dataset. Meanwhile, Table 8 presents the average values and standard deviation after five runs of performance metrics of the approach, utilizing both the augmented and original wildfire datasets.

**Table 7.** Hyperparameter configuration for Method 2.

| Hyperparameter | Value |
| --- | --- |
| Learning Rate | $1 \times 10^{-2}$ |
| Optimizer | Adam |
| Batch Size | 32 |
| Epochs | 30 |
| Fine tune at the layer number | 1 |

**Table 8.** Performance metrics for Method 2 using augmented and original wildfire datasets.

| Metrics | Augmented Wildfire Dataset | Original Wildfire Dataset |
| --- | --- | --- |
| Primary Accuracy | Mean = 0.8766, Std = 0.0147 | Mean = 0.9073, Std = 0.0077 |
| Precision | Mean = 0.7974, Std = 0.0206 | Mean = 0.8832, Std = 0.0263 |
| Recall | Mean = 0.9171, Std = 0.0273 | Mean = 0.8855, Std = 0.0247 |
| F1-Score | Mean = 0.8526, Std = 0.0146 | Mean = 0.8839, Std = 0.0201 |
| ROC-AUC | Mean = 0.8842, Std = 0.0127 | Mean = 0.9053, Std = 0.0164 |

These results are based on five runs for each model, with bootstrapping used to calculate 95% confidence intervals for the differences in performance metrics between the two models.

Data Balancing Assessment for Method 2

The comparison between Method 2 trained on the original wildfire dataset and the augmented dataset reveals nuanced differences in performance across various metrics. The mean primary accuracy for the augmented dataset is 0.8766 (Std = 0.0147), whereas the original dataset yields a slightly higher mean accuracy of 0.9073 (Std = 0.0077), with a significant difference detected within a 95% confidence interval of [0.04396, 0.0166]. Similar to Method 1, the augmented method shows significantly lower precision but higher recall. Again, given the higher number of nofire instances in the test set, the original dataset's performance, particularly its higher precision and overall accuracy, may be attributed to a bias arising from the training data imbalance. In contrast, the augmented dataset method, with its higher recall, appears to offer advantages in sensitivity to fire instances, potentially providing more robustness to class imbalance.

The trade-offs between these two methods extend to the F1-Score and ROC-AUC metrics, with the augmented method generally underperforming. These findings highlight the delicate balance between various evaluation metrics and the importance of alignment between the model's objectives and its evaluation criteria. Specifically, while augmenting positive fire instances enhances sensitivity (recall), it appears to do so at the cost of precision, accuracy, F1-Score, and ROC-AUC. This trade-off underscores the importance of data balance and augmentation strategy and calls for meticulous evaluation and design to achieve the desired model performance in forest fire detection.

Again, these insights stress the need for a comprehensive approach in selecting and fine-tuning models, taking into account not only performance metrics but also the inherent characteristics and potential biases of the datasets being used.

### 3.3. Comparison between Method 1 and Method 2

The comparative analysis of Method 1, a conventional one-step binary classification approach, and Method 2, a hierarchical multi-class classification strategy within a multi-task learning framework, provides an insightful exploration of the strengths and trade-offs between the two approaches in the context of wildfire detection. Refer to Table 9 for a comparison of the two methods.

**Table 9.** Performance metrics for Method 1 and 2 using the augmented wildfire datasets.

| Metric | Method 1 | Method 2 |
|---|---|---|
| Primary Accuracy | Mean = 0.8156, Std = 0.0152 | Mean = 0.8766, Std = 0.0147 |
| Precision | Mean = 0.7187, Std = 0.0143 | Mean = 0.7974, Std = 0.0206 |
| Recall | Mean = 0.8667, Std = 0.0108 | Mean = 0.9171, Std = 0.0273 |
| F1-Score | Mean = 0.7857, Std = 0.0103 | Mean = 0.8526, Std = 0.0146 |
| ROC-AUC | Mean = 0.8258, Std = 0.0083 | Mean = 0.8842, Std = 0.0127 |

These results are based on five runs for each model, with bootstrapping used to calculate 95% confidence intervals for the differences in performance metrics between the two methods.

Primary Accuracy:

- Method 1: Achieves a mean accuracy of 0.8156 with a standard deviation of 0.0142.
- Method 2: Surpasses Method 1 with a mean accuracy of 0.8766 and a standard deviation of 0.0130.
- 95% confidence interval for the difference: [0.05270, 0.06926].
- Discussion: Method 2's approach, which incorporates the simultaneous classification into five distinct subclasses, appears to enhance overall accuracy. This significant improvement may be attributed to the model's refined ability to discriminate between different visual aspects of fire events, such as separating smoke from flames.

Precision:

- Method 1: Records a mean precision of 0.7187 with a standard deviation of 0.0140.
- Method 2: Outperforms with a mean precision of 0.7974 and a standard deviation of 0.0207.
- 95% confidence interval for the difference: [0.05814, 0.09942].
- Discussion: Higher precision in Method 2 suggests that the multi-classification strategy is more effective at correctly identifying true fire instances. The significant differentiation between fire confounding/smoke confounding elements and fire/smoke may contribute to this improvement.

Recall:

- Method 1: Registers a mean recall of 0.8667 with a standard deviation of 0.0128.
- Method 2: Further improves recall with a mean of 0.9170 and a standard deviation of 0.0268.
- 95% confidence interval for the difference: [0.02390, 0.07301].
- Discussion: The significant increase in recall for Method 2 underscores its sensitivity in detecting fire instances. This demonstrates the robustness of the MTL approach.

F1-Score:

- Method 1: Achieves a mean F1-Score of 0.7857 with a standard deviation of 0.0095.
- Method 2: Excels with a mean F1-Score of 0.8526 and a standard deviation of 0.0147.
- 95% confidence interval for the difference: [0.05810, 0.07556].
- Discussion: The higher and significant F1-Score in Method 2 reflects a balanced performance in terms of precision and recall. This confirms that the multi-class MTL-based classification approach is more harmonious in balancing false positives and false negatives.

ROC-AUC:

- Method 1: Records a mean ROC-AUC of 0.8258 with a standard deviation of 0.0086.
- Method 2: Outperforms with a mean ROC-AUC of 0.8844 and a standard deviation of 0.0128.
- 95% confidence interval for the difference: [0.05094, 0.06582].
- Discussion: The significant and superior ROC-AUC score in Method 2 indicates that it performs better in ranking predictions and maintaining a favorable trade-off between the true positive rate and false positive rate.

Conclusion: The results highlight the efficacy of Method 2′s hierarchical multi-class classification strategy. By simultaneously handling multiple classification tasks and considering the nuances of various fire and nofire-related elements, it demonstrates statistically significant improvements across all the considered metrics compared to the traditional one-step approach of Method 1. The integration of confounding elements within the MTL framework appears to offer a more robust modeling of wildfire events.

### 3.4. The Evaluation on the Wildfire Dataset

The evaluation of the newly introduced wildfire dataset, conducted against a combined set of previously published datasets totaling 36,775 images, provides critical insights into the performance and adaptability of our model. Despite the comprehensive nature and extensive range of the combined dataset, the model's accuracy on the wildfire dataset reached 0.7936.

### 3.5. Analysis of Weight Sensitivity on Confounding Elements Subclasses

The analysis of weight sensitivity on confounding elements subclasses is conducted to assess how varying weights influence the model's performance metrics. This investigation provides insights into the model's responsiveness to the considered distinct characteristics of fire and nofire instances. Table 10 shows the results of model performance with varying weights through multiple combinations in the range of {1, 2, 3} × {1, 2, 3} for the two subclasses related to the confounding elements (subclass 4 and 5). Each value represents the computed mean after five runs.

**Table 10.** Results of model performance metrics means after five independents runs with varying weights for confounding elements subclasses.

| Weight Combination (Subclass 1, Subclass 2, Subclass 3, Subclass 4, Subclass 5) | Primary Accuracy | Precision | Recall | F1-Score | ROC-AUC |
|---|---|---|---|---|---|
| (1.0, 1.0, 1.0, 0.2, 0.3) | 0.9098 | 0.8115 | 0.9748 | 0.8857 | 0.9157 |
| (1.0, 1.0, 1.0, 0.2, 0.2) | 0.8716 | 0.7804 | 0.9392 | 0.8524 | 0.8859 |
| (1.0, 1.0, 1.0, 0.1, 0.3) | 0.9015 | 0.8348 | 0.9207 | 0.8756 | 0.9026 |
| (1.0, 1.0, 1.0, 0.3, 0.1) | 0.8683 | 0.8212 | 0.9245 | 0.8698 | 0.8985 |
| (1.0, 1.0, 1.0, 0.3, 0.2) | 0.8878 | 0.8212 | 0.9245 | 0.8698 | 0.8985 |
| (1.0, 1.0, 1.0, 0.2, 0.1) | 0.8829 | 0.7937 | 0.9434 | 0.8621 | 0.8940 |
| (1.0, 1.0, 1.0, 0.1, 0.2) | 0.9024 | 0.8599 | 0.8491 | 0.8544 | 0.8807 |
| (1.0, 1.0, 1.0, 0.3, 0.3) | 0.9024 | 0.7979 | 0.9686 | 0.8750 | 0.9066 |

The weight combination (subclass 1, subclass 2, subclass 3, subclass 4, subclass 5) corresponding to (1.0, 1.0, 1.0, 0.2, 0.3) signifies that all the first three subclasses are assigned a weight of one, while the 4th and 5th subclasses, which represent the fire confounding elements and the smoke confounding elements, are given respective weights of 2 and 3.

#### 3.5.1. Sensitivity to Fire Confounding Elements

Increasing the weight of fire confounding elements (e.g., the weight combination [1.0, 1.0, 1.0, 0.3, 0.1]) generally leads to higher precision and a slight decrease in recall. This trend indicates that the model becomes more selective in identifying true fire instances, potentially missing some challenging cases related to smoke instances (subclasses 1 and 5). This pattern may underscore the relatively greater importance of smoke-related instances as opposed to fire ones (subclasses 2 and 4), possibly reflecting the higher number of smoke-related instances in the test dataset. This trend might be observable in other combinations, though the specific combination (1.0, 1.0, 1.0, 0.2, 0.1) appears to be an exception and might not fully align with this general observation.

#### 3.5.2. Sensitivity to Smoke-Confounding Elements

The weighting of smoke confounding elements demonstrates noticeable effects on the model's performance. Specifically, reducing the weight on these elements (e.g., [1.0, 1.0,

1.0, 0.1, 0.2]) leads to higher precision and lower recall. This reflects a more conservative classification approach towards smoke detection, suggesting a nuanced sensitivity to these elements. Again, this pattern may underscore the relatively greater importance of smoke-related instances as opposed to fire ones (subclasses 2 and 4), possibly reflecting the higher number of smoke-related instances in the test dataset.

### 3.5.3. Equal Weights and Balanced Performance

When equal weights are assigned to both fire confounding elements and smoke confounding elements (e.g., [1.0, 1.0, 1.0, 0.3, 0.3]), the model produces relatively balanced performance across all key metrics. This observation highlights the model's well-rounded responsiveness to various confounding elements, fostering balanced detection.

### 3.5.4. Stability in Primary Accuracy

A noteworthy consistency is found in primary accuracy, ranging from 0.8683 to 0.9098 across different weight combinations. This stability indicates that the model maintains its overall classification ability while fine-tuning its sensitivity to the confounding elements.

### 3.5.5. Influence of Instances Distribution

The distribution of instances, particularly the fewer instances of fire confounding elements, underscores the value of weighting. By compensating for this disparity, the approach may show some potential combinations that learn more robustly from this subclass, illustrating the importance of weighting in achieving balanced learning and the potential that offers the wildfire's dataset unique structure.

### 3.5.6. Conclusion

Overall, the analysis reveals intricate tendencies in the model's response to varying weights for confounding elements. The trade-off between precision and recall, the stability in primary accuracy, and the influence of instances distribution reflect a multifaceted interaction. The results imply that there may not be a one-size-fits-all approach to weighting, but a systematic exploration can guide the identification of an optimal balance. Such balance aligns with the specific goals and constraints of fire/smoke detection applications, contributing to a more versatile model.

## 4. Discussion

Upon an in-depth analysis of deep learning forest fire models using RGB images, the existing body of literature reveals a noticeable gap—a comprehensive scrutiny of data representativeness and challenging factors that can induce high false alarm rates. While several studies do acknowledge the correlation between these confounding elements and false alarms, their treatment of the subject often lacks the breadth and depth necessary for a complete understanding.

This research endeavor addresses this specific gap in several ways. First, it crafts a more exhaustive enumeration of these confounding elements. Second, and more importantly, it introduces the wildfire dataset, a significant contribution that effectively considers these confounding elements. This dataset offers researchers a valuable tool for data collection, annotation, and model performance evaluation, thereby aiding in a more nuanced understanding and effective management of the factors that lead to false alarms in forest fire detection.

The evaluation of the newly introduced wildfire dataset, conducted against a combined set of previously published datasets totaling 36,775 images, yields critical insights. Despite the comprehensive nature and extensive range of the combined dataset, the model's accuracy on the wildfire dataset reached 0.7936. Given the considerable size of the training set, this accuracy might appear moderate. A subjective error analysis further bolstered indicates that the wildfire dataset introduces a unique set of challenges and confounding elements that may not have been sufficiently represented in the previously used datasets.

Moreover, in the context of forest fire detection, it is vital to highlight that the task at hand is fundamentally a binary classification problem, distinguishing between fire and nofire events. The ostensibly straightforward nature of the problem might suggest that achieving high accuracy should be less challenging. However, when the model was exposed to the wildfire dataset, an accuracy of 0.7936 was obtained, despite the substantial volume of the combined dataset used for training. This underscores the complexity of the task and the importance of carefully considering the confounding elements specific to forest fire detection and in general, data representativeness.

This performance, while respectable in some contexts, bears significant implications when translated into real-world, practical applications of forest fire detection. Given the high stakes associated with accurate and timely forest fire detection, where errors could result in substantial environmental damage and potential loss of life, an accuracy rate below 100% is of critical concern. This underscores the complexity of real-world forest fire detection tasks, which must contend with an array of confounding elements not adequately represented in existing datasets. The results also emphasize the importance of our endeavor in creating the wildfire dataset, which introduces new challenging elements, making it a valuable tool in the development of more robust and reliable DL models for forest fire detection. The results, therefore, validate the significance of our study's contribution in this field.

Further, the present work delves into the intricate issue of data balancing, especially pertinent in the context of wildfire detection, where imbalances between fire and nofire classes are intrinsic. An augmented dataset was meticulously crafted by employing a nuanced method to equalize the fire training class with the nofire class, leaving the validation and test datasets untouched. The empirical comparison between the original and augmented training sets unveiled subtle yet essential insights into the influence of data balancing on model performance. While the original dataset resulted in higher accuracy, precision, recall, F1-score, and ROC-AUC, this study's comprehensive exploration reveals that these metrics, in isolation, may not sufficiently capture the model's effectiveness, particularly in the complex context of imbalanced data. The greater accuracy in the original dataset could be misleading, as a model may overly favor the majority class. A judicious examination of this pattern has led to a broader understanding of the delicate interplay between statistical representation, real-world occurrence, and the model's intended role. By adopting a thoughtful and strategic approach to data balancing, this study accentuates the necessity of a measured approach that transcends mere numerical evaluation, contemplating the underlying data dynamics, the practical complexity of wildfire detection, and the overarching goal of crafting models attuned to sophisticated, real-world applications.

Another significant facet of this study is the introduction of an innovative multi-task learning approach for forest fire detection. This approach introduces an auxiliary task of five-class classification trained simultaneously with the primary binary classification task (fire vs. nofire). A single base model is employed to manage both tasks concurrently, exploiting the shared features between tasks to enhance the model's ability to generalize and improve its overall performance. The study undertakes a comparative assessment between this hierarchical multi-class classification strategy (Method 2) and a traditional one-step binary classification approach (Method 1). In the one-step approach, the same model is trained directly to categorize images into two general classes: images showcasing a fire event (classes 1 and 2) and those that do not (classes 3, 4, and 5). The intention behind this analysis is to shed light on the potential benefits of the multi-class, MTL-based classification strategy and underline the potential of addressing confounding elements within the method as independent subclasses (classes 4 and 5).

When evaluated on the augmented wildfire dataset, the two-step multi-class classification approach (Method 2) showcased a significant suitability over the one-step classification approach (Method 1). The results consistently favor the latter across all key metrics including accuracy, precision, recall, F1-Score, and ROC-AUC. Method 2 demonstrates a mean accuracy of 0.8766 and an F1-Score of 0.8526, reflecting a refined ability to distinguish

different visual aspects of fire events, such as smoke from flames, and excels in ranking predictions. The significant increases in these metrics and the corresponding 95% confidence intervals ([0.05270, 0.06926] for accuracy, [0.05814, 0.09942] for precision, [0.02390, 0.07301] for recall, [0.05810, 0.07556] for F1-Score, and [0.05094, 0.06582] for ROC-AUC) underline the robustness and nuanced modeling of wildfire events through Method 2. This comparison emphasizes the efficacy of a novel methodological design, with the integration of confounding elements within the multi-task learning framework, to attain significant improvements in wildfire detection performance.

A subjective error analysis using feature visualization techniques, such as Grad-CAM, further bolstered the superiority of Method 2, pointing out its advantage not only in terms of standard performance metrics, but also in its ability to more accurately discern the subtleties of the images within the dataset.

In moving forward, the uniquely structured wildfire dataset introduced in this study presents promising opportunities for further research in wildfire detection. Its distinct subclasses and the careful consideration of environmental and caption-related variability provide a multifaceted platform that may contribute to the refinement of detection algorithms and the advancement of image processing techniques. Building upon the insights gathered from this study, the wildfire dataset, with its nuanced structure and five distinct subclasses, offers promising avenues for further research and practical applications. The inclusion of diverse categories such as smoke from fires, both smoke and fire, forested areas, and confounding elements adds layers of complexity that can facilitate more targeted and nuanced analyses.

The specific subclass categorization may provide researchers with a platform to investigate specialized aspects of wildfire detection, such as distinguishing smoke from fire, recognizing elements that can be misidentified as fire or smoke, or understanding the interplay between flames and smoke. This can lead to more refined algorithm development, though further validation and experimentation would be necessary to confirm these possibilities.

Furthermore, the dataset's attention to environmental variability, including aspects such as topography, forest types, weather conditions, and fire characteristics, opens up opportunities for modeling different wildfire scenarios. This could aid in creating more adaptable detection models, enhancing their relevance to diverse real-world contexts, without asserting that it can completely overcome all challenges.

In terms of image processing, the caption-related variability captured in the dataset presents a valuable resource for researchers. Factors such as lighting conditions, image resolution, altitude, and angle offer a range of challenges that may contribute to the development of techniques for improving image quality and robustness under various conditions.

Additionally, the dataset may serve as a potential benchmark for evaluating and comparing different wildfire detection models. Its rich structure and variety could provide a basis for assessing algorithm performance, though it would need to be utilized within a controlled experimental setup to ensure fair comparisons.

Furthermore, the wildfire dataset distinct structure, especially the inclusion of subclasses representing various confounding elements, allows for the utilization of advanced feature visualization techniques, such as Grad-CAM. By offering these insights, researchers may gain a more detailed understanding of how specific regions of the images influence the model's predictions. This has the potential to uncover hidden patterns and dynamics that models trained on conventional datasets might overlook. Furthermore, the subclasses can enable a more granular examination of feature maps within visually similar images that have contrasting classification outcomes. Identifying connections between particular visual elements and instances of confusion in the predictions becomes an attainable goal. Though promising, the realization of this potential may require careful handling, as the complexity of the confounding elements may present unforeseen challenges.

An ablation study may be another avenue of exploration, wherein the removal of specific features or sections of the model can help assess their impact on performance. The

systematic analysis might unveil areas causing confusion and provide data-driven insights to guide further data collection or architecture adjustments. This can be a vital step towards more accurate forest fire detection, though it must be undertaken with due consideration to the multifaceted nature of the data and models.

By taking advantage of the wildfire dataset structure, it provides a scaffold for innovation that might lead to the development of thoughtful and methodical implementation, recognizing that the intricate dynamics of the data can offer both opportunities and challenges. In this context, one promising avenue is the possibility to permit a novel methodological design, such as for example, the integration of confounding elements within the multi-task learning framework. This approach (Method 2) showed significant improvements by leveraging the diverse characteristics found within the dataset, utilizing the nuances in image categorization to create more responsive models. In that sense, varying the weights of these subclasses showed potential avenues of improvement as reported in Section 3.5.

In summary, the dataset's careful attention to detail and classification offers fertile grounds for ongoing innovation in wildfire detection. Whether through advanced visualization, novel methodological designs, or nuanced weighting strategies, the possibilities are vast and compelling. However, each step forward also demands a measured understanding of the underlying dynamics and a willingness to adapt and refine methods as new insights emerge. The ultimate goal remains clear: to foster models capable of sophisticated, real-world applications, making strides towards a more effective and comprehensive approach to wildfire detection and management.

*Study Limitations*

One notable limitation of the present study is the absence of external validation using a completely independent dataset, not involved in any phase of the model development process. While the design and execution of the experiments were meticulous, they were conducted exclusively on the curated dataset. This focus could potentially constrain the generalizability of the findings, limiting their applicability to other regions or different wildfire scenarios. External validation with diverse and unrelated data would have provided a more rigorous test of the model's robustness and its ability to adapt to variations beyond the characteristics captured in the training and validation sets. Future research efforts that include such external validation can further substantiate the approaches's efficacy and contribute to a more comprehensive understanding of its performance in real-world wildfire detection and management.

Another limitation of the study is the lack of consideration for the essential factors of time and computational resources in the methodology. While these aspects were considered in the choice of the model, they were not explicitly integrated into the evaluation of the dataset's structure or the effects of various methodological approaches. The time required for training, validating, and testing the models, as well as the computational resources necessary for these processes, plays a vital role in the practical applicability of the findings. Ignoring these aspects may lead to an incomplete understanding of the model's efficiency and feasibility in real-world scenarios.

These limitations do not diminish the value of the study but rather highlight areas for further refinement and exploration. Future research should aim to address these issues by developing more objective and standardized classification criteria for the subclasses, as well as by incorporating a more comprehensive assessment of time and computational demands in the methodology. By acknowledging and addressing these limitations, subsequent studies can build upon the current findings, contributing to a more robust and nuanced understanding of wildfire detection and the effective utilization of the novel dataset.

## 5. Conclusions

The potential to bridge the existing divide between the theoretical algorithms discussed in the literature and their practical applications becomes a paramount objective for future research endeavors. Prioritizing the enhancement of the representativeness

in training and testing datasets emerges as a vital step in this direction. By doing so, it becomes feasible to develop more accurate and robust forest fire detection systems capable of effectively handling the diverse complexities encountered in real-world scenarios. This focus serves not only as the next logical step in this line of research but also as a crucial contribution towards the broader goal of creating more reliable and efficient solutions for forest fire detection.

**Supplementary Materials:** The wildfire dataset is made available for public access. Accompanying this dataset is a CSV document that provides a direct download URL for each image, details its resolution, and includes relevant metadata. Access Link: https://kaggle.com/datasets/elmadafri/the-wildfire-dataset. Researchers are encouraged to use this resource for their studies and to cite appropriately when used.

**Author Contributions:** Conceptualization, I.E.-M., M.P. and N.O.-T.; Methodology, I.E.-M., M.P. and N.O.-T.; Software, I.E.-M.; Validation, I.E.-M., M.P. and N.O.-T.; Formal analysis, I.E.-M., M.P. and N.O.-T.; Investigation, I.E.-M., M.P. and N.O.-T.; Data curation, I.E.-M.; Writing—original draft, I.E.-M.; Writing—review & editing, M.P. and N.O.-T.; Visualization, I.E.-M.; Supervision, M.P. and N.O.-T.; Project administration, M.P. and N.O.-T. All authors have read and agreed to the published version of the manuscript.

**Funding:** This research received no external funding.

**Informed Consent Statement:** The authors of the dataset are committed to ensuring that the use of the images within the dataset adheres to the respective licensing requirements and permissions. In the event that an image owner has concerns regarding the inclusion of their image or believes that their copyright or licensing rights have been infringed, the authors are open to addressing these concerns and replacing the image in question with an alternative that meets the dataset's criteria. This approach aims to maintain a respectful and collaborative relationship with image owners and the broader research community while upholding the dataset's integrity and legal compliance.

**Data Availability Statement:** All datasets and code necessary to corroborate the present research findings are available at the following link: https://kaggle.com/datasets/elmadafri/the-wildfire-dataset. Researchers are encouraged to use these resources for further studies.

**Acknowledgments:** The authors would like to acknowledge the assistance of AI-based language models in refining the language quality and coherence of the manuscript. This AI tool has been instrumental in formulating clear and concise scientific statements, facilitating more effective communication of our research results and contributions.

**Conflicts of Interest:** The authors declare that they have no known competing financial interests or personal relationships that could have appeared to influence the work reported in this paper.

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
