# Peer review of "The Wildfire Dataset: Enhancing Deep Learning-Based Forest Fire Detection with a Diverse Evolving Open-Source Dataset Focused on Data Representativeness and a Novel Multi-Task Learning Approach"

_forests, doi:10.3390/f14091697_

Round 1

Reviewer 1 Report

The paper introduces a comprehensive dataset called the wildfire dataset, which captures a wide variety of environmental conditions, forest types, geographical regions, and confounding elements. The authors propose a novel Multi-Task Learning (MTL) approach that integrates multi-class confounding elements within the framework. The paper provides a detailed description of the dataset collection and curation process, model selection, training methodology, and performance evaluation. The results show that the MTL approach outperforms the traditional one-step classification approach, and the wildfire dataset demonstrates its potential in improving the accuracy and reliability of forest fire detection models. Overall, the paper addresses an important problem in the field of forest fire detection.

Some suggestions for improvement of this paper are as follows:

1. Provide more details about the wildfire dataset: The manuscript briefly mentions that the wildfire dataset comprises 2700 RGB instances and captures a wide range of environmental conditions, forest types, geographical regions, and confounding elements. It would be beneficial to provide more specific details about the dataset, such as the distribution of images across classes and subclasses and any additional preprocessing steps applied to ensure the dataset's integrity.

2. Data balancing can help improve the performance of deep learning models by avoiding overfitting to a particular category or neglecting others. This can be added.

3. The manuscript does not address the weight of confounding elements. Different confounding elements may have varying impacts on the model's performance. It is suggested to incorporate weighting of confounding elements in the experiments to assess the model's performance more accurately under different confounding elements.

4. Include more information about the evaluation metrics: The manuscript mentions the evaluation metrics used, such as accuracy, precision, recall, and F1-score. However, it would be useful to include additional performance metrics, such as the area under the receiver operating characteristic curve (AUC-ROC), to provide a more comprehensive evaluation of the models' performance.

5. It would be beneficial to include the values of these metrics for both Method 1 and Method 2 in a table or figure, allowing for a direct comparison of their performance.

6. Discuss the potential applications: Discussing potential applications of the wildfire dataset, such as its use in training and evaluating other fire detection models, would further emphasize its value to the research community.

7. Discuss the limitations of the study: It would be valuable to include a section discussing the limitations of the study. For example, the study could address potential challenges in generalizing the results to different geographic regions or the limitations of using RGB images for forest fire detection. Discussing these limitations will provide a balanced perspective and guide future research directions.

The paper is well-written, and the language is clear and concise. The authors effectively communicate their ideas and findings. However, there are a few minor grammatical errors and typos that should be corrected during the revision process.

Author Response

Response to Reviewer 1 Comments

Dear Referee,

We would like to thank you for the constructive comments and suggestions provided for the manuscript titled "The Wildfire Dataset: Enhancing Deep Learning-Based Forest Fire Detection with a Diverse Evolving Open-Source Dataset Focused on Data representativeness and a Novel Multi-Task Learning approach." Your insights have greatly helped in enhancing the quality and clarity of our work.

In the revised version of the manuscript, all the modified, added, and revised relevant parts have been highlighted with a light blue background to facilitate your review. This way, you can easily identify the changes and assess how they align with your comments.

Now, we would like to address each of your points as follows:

Comments on the Quality of English Language: The paper is well-written, and the language is clear and concise. The authors effectively communicate their ideas and findings. However, there are a few minor grammatical errors and typos that should be corrected during the revision process.

We would like to express our gratitude for the commentary provided on the language quality of our original manuscript. In response to your suggestions, we have meticulously reviewed and made substantial improvements to the English language in the revised version of this work. This effort was undertaken to enhance clarity, coherence, and readability, ensuring that our research findings are presented in a clear and scientific manner. We believe that these revisions have successfully addressed the concerns raised and hope that the updated manuscript meets your expectations.

Point 1: Provide more details about the wildfire dataset: The manuscript briefly mentions that the wildfire dataset comprises 2700 RGB instances and captures a wide range of environmental conditions, forest types, geographical regions, and confounding elements. It would be beneficial to provide more specific details about the dataset, such as the distribution of images across classes and subclasses and any additional preprocessing steps applied to ensure the dataset's integrity.

Response 1: Please provide your response for Point 1. (in red)

In response to your suggestion, we have enriched the manuscript with more specific information regarding the dataset’s structure.

  1. Distribution and Structure: We have introduced relevant information on the partitioning of the dataset into training, validation, and testing subsets, adhering to a 70-15-15% split ratio. We have also detailed the distribution of images across the 'fire' and 'nofire' classes and their respective subclasses. An enhanced description of the training, validation, and testing sets has been included, detailing their structure and utilization in model learning, fine-tuning, and evaluation throughout the manuscript. However, this extension is mainly reflected in the '2.1.1. Image Collection and Dataset Structure Formation' section of the manuscript.

  1. Detailed Table 2: Table 2 has been added to the manuscript that precisely outlines the distribution of instances across subclasses for each set, providing a clear view of the dataset's composition.

  1. Randomization: We have described the process of randomizing the images within each directory (training, validation and test), ensuring a diverse representation across the dataset, thus preserving its integrity.

  1. Preprocessing Considerations: No preprocessing steps were applied to the images to make them more versatile and usable in various contexts. However, preprocessing steps tailored to the objectives of the present work were applied in the context of the study's experiments, explained in section 2.3. 

We hope this general overview of the enhancements made to the manuscript clarifies the dataset's structure, distribution, and management. We believe that these additions will provide readers with a better understanding of the dataset's composition and its application in our study.

Point 2: Data balancing can help improve the performance of deep learning models by avoiding overfitting to a particular category or neglecting others. This can be added.

Response 2: Please provide your response for Point 2. (in red)

Thank you for highlighting the significance of data balancing in our manuscript. Your valuable feedback has sparked a thorough examination and update of our approach to this essential aspect of applied deep learning. Consequently, new sections have been added:

Section '2.6. The Data Balancing Problem' was introduced to explain the challenges of and need for data balancing in the wildfire dataset, due to unequal class distributions. This section detailed the methodology applied to augment only the fire class within the training set, using techniques like random rotations and flipping, and ensured that no source images were duplicated in the augmentation process. The considerations guiding this decision included avoiding overfitting, preserving natural distribution in validation and test sets, and preventing data leakage.

Empirical comparisons were made between the original and augmented datasets, analyzing differences in key performance metrics. This comparison, based on two methods, One-Step Classification (Method 1) and Two-Step MTL Multi-Class Classification (Method 2), helped in understanding the complex relationship between data balance and model efficacy. The impact analysis revealed that while there were differences (with higher results when the model is trained on the original training set), the balanced dataset maintained robust results, possibly reflecting a more nuanced model. 

Further, sections '3.1.1. Data Balancing Assessment for Method 1' and '3.2.1. Data Balancing Assessment for Method 2' were added to provide insights into the model's capacity to classify and discriminate between classes.

Tables (3,6,8) were introduced to represent the distribution instances of the balanced dataset and the results obtained from the comparison for each method. 

The 'Revised Discussion Section' delved into the insights gained from the balancing assessment that have helped present a discussion that reflects real-world applicability challenges in applying RGB-based deep learning models to forest fire detection.

All subsequent experiments were trained on the balanced (augmented) training set.

 Point 3: The manuscript does not address the weight of confounding elements. Different confounding elements may have varying impacts on the model's performance. It is suggested to incorporate weighting of confounding elements in the experiments to assess the model's performance more accurately under different confounding elements.

Response 3: Please provide your response for Point 3. (in red)

We wholeheartedly appreciate your thoughtful suggestion on the necessity of considering the weight of confounding elements in assessing the model's performance. In response, we have extensively augmented the manuscript to present a comprehensive analysis of this important aspect. Consequently, new sections have been added:

Section 2.7 “ Weighting of Confounding Elements subclasses in Model Training”:

 We introduced a systematic approach by varying the weights of these two subclasses (subclasses 4 and 5) incrementally, from all the combinations in {1,2,3}×{1,2,3}, we then analyzed how the model's performance responds to different confounding elements scenarios. This analysis extends to a discussion of the underlying mechanisms that make certain weightings effective.

Section 3.5 “Analysis of Weight Sensitivity on Confounding Elements Subclasses”:

 We carried out a detailed study of how varying weights influence a model's key metrics. The findings include:

  • Sensitivity to Fire-Confounding Elements: Exploring the specificity in identifying true fire instances and its effects on precision and recall.
  • Sensitivity to Smoke-Confounding Elements: Understanding the model's nuanced sensitivity towards smoke detection.
  • Equal Weights and Balanced Performance: Observing balanced detection across metrics when equal weights are assigned.
  • Stability in Primary Accuracy: Noticing quite consistent overall classification ability across different weight combinations.
  • Influence of Instances Distribution: Emphasizing the importance of weighting in achieving balanced learning considering disparities in instances.

Table 10:

 The table shows the results of model performance with varying weights through multiple combinations in the range of {1,2,3}×{1,2,3} for the two subclasses (4 and 5) related to the confounding elements. Each value is the computed mean after five runs. 

“Revised Discussion Section”

 The revised manuscript’s discussion  section reflects a multifaceted understanding of weighting, recognizing the intricacies in the model's response to varying weights and the need for a tailored approach to balance precision, recall, and other metrics. Indeed, while the variations between the datasets might suggest differences in model performance, they emphasize a more nuanced, realistic, and generalized modeling approach. The results, though distinct across the two datasets, underscore the intricate connection between data balancing strategies, the model's purpose, the underlying data structure, and real-world class dynamics. This insight illuminates the value of a multifaceted evaluation.

Overall, the newly introduced content comprehensively addresses the concern, showing the potential of considering confounding elements in fire detection. We believe that these revisions enhance the manuscript's depth, providing valuable insights into the complex interplay of factors affecting model performance.

Point 4: Include more information about the evaluation metrics: The manuscript mentions the evaluation metrics used, such as accuracy, precision, recall, and F1-score. However, it would be useful to include additional performance metrics, such as the area under the receiver operating characteristic curve (AUC-ROC), to provide a more comprehensive evaluation of the models' performance.

Response 4: Please provide your response for Point 4. (in red)

Thank you for pointing out the importance of a comprehensive evaluation of the models' performance. We agree that the ROC-AUC score provides additional valuable insights into the discriminatory power of the models. The ROC-AUC score was then introduced and considered in our study.

As described in Section 2.9, "Performance Metrics," the ROC-AUC (Receiver Operating Characteristic - Area Under Curve) score was defined and its significance in evaluating a model's ability to distinguish between classes was detailed. This metric was subsequently employed in all of the study's experiments, with the exception of Section 3.4, "The Evaluation on the Wildfire Dataset." The inclusion of ROC-AUC, alongside accuracy, precision, recall, and F1-score, ensures a nuanced and robust assessment of the models, contributing to the integrity and comprehensiveness of our analysis.

  1. It would be beneficial to include the values of these metrics for both Method 1 and Method 2 in a table or figure, allowing for a direct comparison of their performance.

Response 5: Please provide your response for Point 5. (in red)

Thank you for emphasizing the importance of a clear and direct interpretation of the comparison between Method 1 and Method 2. We agree that presenting the values of the performance metrics in a visually accessible way enhances the interpretability of our results.

We would like to inform you that the comparison between Method 1 and Method 2, including the values for accuracy, precision, recall, F1-score, and ROC-AUC, has been organized into Table 9 in Section 3.3, "Comparison between Method 1 and Method 2." . 

Additionally, we adopted a similar presentation strategy throughout the rest of the study's experiments to ensure consistency and enhance visibility. This has allowed an easier and effective comparison across various analyses.

  1. Discuss the potential applications: Discussing potential applications of the wildfire dataset, such as its use in training and evaluating other fire detection models, would further emphasize its value to the research community.

Response 6: Please provide your response for Point 6. (in red)

We are grateful for the insightful suggestion to discuss the potential applications of the wildfire dataset. In response, we have expanded the discussion in the revised discussion section to elaborate on the diverse opportunities that the dataset provides for further research and practical applications in wildfire detection. These include the possibility of refining detection algorithms, advancing image processing techniques, and serving as a benchmark for evaluating different models. Furthermore, the intricate design of the dataset allows for innovative exploration in recognizing various fire characteristics and confounding elements, fostering the development of more context-aware models. We believe these insights emphasize the value of the dataset to the research community, aligning with your valuable recommendation.

  1. Discuss the limitations of the study: It would be valuable to include a section discussing the limitations of the study. For example, the study could address potential challenges in generalizing the results to different geographic regions or the limitations of using RGB images for forest fire detection. Discussing these limitations will provide a balanced perspective and guide future research directions.

Response 7: Please provide your response for Point 7 (in red)

We appreciate the valuable commentary and have addressed it by introducing a new section “4.1 Study limitations” within the discussion section to elaborate on the limitations of our study. 

Addressing RGB-based data-driven models limitations is highly relevant. In this sense, the introduction’s manuscript highlights the inherent limitations of using RGB data alone for deep learning-based forest fire detection. It emphasizes the necessity of integrating RGB data within a multi-modal approach, which leverages different data modalities in a complementary way to take advantage of their combined strengths.

On the other hand, the introduced section covers the absence of external validation, which may affect the generalizability of our findings to different regions or wildfire scenarios, and also, the lack of consideration for time and computational resources. These acknowledgments offer a balanced view and will indeed guide future research directions. 

Once again, thank you for your insightful feedback, which has undoubtedly improved the clarity and comprehensiveness of our manuscript.

Reviewer 2 Report

It is very interesting in reviewing this manuscript. I am especially interested by the ways which you used to structure your dataset. The analyses you conducted to subclasses of confounding elements provide a good opportunity for a deeper exploration and understanding of the complicated dynamics of forest fire detection models. Your analytical results and the methods used contribute to our development of more effective and reliable models for detecting forest fires.

Use of your wildfire dataset illustrated higher accuracy and lower false alarm rates than a traditional binary classification method. These publicly acceptable data make the data set and the modeling methods are more useful practically.

I really enjoyed reading your proposed a novel Multi-Task Learning approach, integrating multi-class confounding elements within the framework. Applying of this strategy in forest fire detection seems indeed may improve the model's discriminatory ability and reduce false positives.

My only suggestion for the authors is that you might want to mention the scales and resolutions of the data set / images you used in your DL exercises, as these are the main factors determine the detecting ability of your models.

Minor editing of English language is needed as a few typos in the manuscript are found, and need to be corrected, notably on, but not limited to, page 1 and page 15.

Author Response

Response to Reviewer 2 Comments

Dear referee, 

We would like to thank you for the constructive comments and suggestions provided for the manuscript titled "The Wildfire Dataset: Enhancing Deep Learning-Based Forest Fire Detection with a Diverse Evolving Open-Source Dataset Focused on Data representativeness and a Novel Multi-Task Learning approach." Your insights have greatly helped in enhancing the quality and clarity of our work.

In the revised version of the manuscript, all the modified, added, and revised parts have been highlighted with a light blue background to facilitate your review. This way, you can easily identify the changes and assess how they align with your comments.

Now, we would like to address each of your points as follows:

Comments on the Quality of English Language: Minor editing of English language is needed as a few typos in the manuscript are found, and need to be corrected, notably on, but not limited to, page 1 and page 15.

We would like to express our gratitude for the commentary provided on the language quality of our original manuscript. In response to your suggestions, we have meticulously reviewed and made substantial improvements to the English language in the revised version of this work. This effort was undertaken to enhance clarity, coherence, and readability, ensuring that our research findings are presented in a clear and scientific manner. We believe that these revisions have successfully addressed the concerns raised and hope that the updated manuscript meets your expectations.

Point 1: My only suggestion for the authors is that you might want to mention the scales and resolutions of the data set / images you used in your DL exercises, as these are the main factors determine the detecting ability of your models. 

Response 1: Please provide your response for Point 1. (in red)

Thank you for your encouraging report and valuable suggestion regarding the scales and resolutions of the images in the introduced dataset. We wholeheartedly agree with your observation that these factors significantly influence the detecting ability of applied deep learning models.

In response to your comment, we have updated the manuscript’s section “2.1.1. Image Collection and Dataset Structure Formation” to include a detailed description of the resolutions and scales associated with the dataset. Specifically, we have provided information about the average, minimum, maximum resolution, and the standard deviation of the resolution for both width and height. The average resolution of 4057×3155 pixels has been highlighted as indicative of high-resolution imagery, which is preferable for precise analysis.

Furthermore, we have also added in the same section information about the diversity in scales represented in the dataset due to varied sources and contexts. This aspect was carefully considered in our deep learning exercises, and the images were resized to a consistent scale, as detailed in section 2.3 of the manuscript.

We have also included the resolution of each individual image of the 2700 images of the wildfire dataset in the CSV document accompanying the dataset. This addition ensures that future researchers utilizing this dataset will have access to detailed information about each image's resolution, enhancing the usability and adaptability of the dataset.

Once again, thank you for your insightful feedback, which has undoubtedly improved the clarity and comprehensiveness of our manuscript.

Round 2

Reviewer 1 Report

From the revised manuscript, it is evident that the authors have fully considered the comments and made adequate revisions accordingly.

The authors provided supplementary details and clarifications on the dataset and processing, addressed the issue of data balancing, which are necessary for readers to better comprehend the research and its rationality. In addition, the authors added further investigation and discussion of the weight of confounding elements, supplemented evaluation metrics for the model, which can improve the completeness and validity of the study. Finally, per suggestions, the authors expanded on the application and limitations of the research in Discussion, which is beneficial for the integrity and reasonability of the paper.

It is clear from these revisions that the authors have taken the feedback into serious consideration. They have made a concerted effort to address all concerns and suggestions, resulting in a more robust and well-supported manuscript.